# Marine Icing Sensor with Phase Discrimination

**DOI:** 10.3390/s21020612

**Published:** 2021-01-17

**Authors:** Abdulrazak Elzaidi, Vlastimil Masek, Stephen Bruneau

**Affiliations:** Faculty of Engineering and Applied Science, Memorial University of Newfoundland, St. John’s, NL A1B 3X5, Canada; masek@mun.ca (V.M.); sbruneau@mun.ca (S.B.)

**Keywords:** electrostatic sensor array, decision tree method, marine icing

## Abstract

In this paper, a novel approach is presented to the measurement of marine icing phenomena under the presence of a two-phase condition. We have developed a sensor consisting of an electrostatic array and a signal processing based on a decision tree method. A three-element electrostatic array is employed to derive signals having linearly decoupled characteristics from which two key parameters, ice and water accretion layer dimension, can be determined for the purpose of environmental monitoring. The quantified characteristics revealed a correlation with the ice layer thickness in spite of the strong influence from the top water phase layer. The decision tree model established a relationship between the signal characteristics and the two accretion thickness parameters of water and ice layer. Through experimental verification, it has been observed that our sensor array in combination with the decision tree model based signal processing provides a simple practical solution to the challenging field of a two phase composition measurement such as in the marine icing considered in this study.

## 1. Introduction

Marine Icing is an adverse phenomena affecting offshore vessels and other structures like wind farms [1] or oil and gas platforms [2]. Icing in general, including the atmospheric icing on airplane wings or transmission lines, poses a great deal of difficulties to many operations and can also be hazardous to personnel in the area especially the operators. The knowledge about the rate of ice growth besides the total accumulation would enable managing these hazardous conditions. One way to mitigate the challenging levels of icing is to heat trace the critical infrastructure or shutting down the operations.

Graz Technical University [3] applied a capacitive tomography to atmospheric icing on high voltage power lines. Graz team considered the two phase phenomena, ice and water, however, due to the focus on the atmospheric icing, the water phase was considered finely dispersed and embedded in the ice phase. We focus on marine icing with a significant water layer on top of the ice.

Combitech IceMonitor [4] measures the ice mass on a rotating rod by a load cell. The system requires a stationary installation which prevents this system being applied to ships and other marine structures due to the dynamic forces, vibration, wind gusts or dynamic water splashes.

The Goodrich 0871LH1 ice detectors [5] use an axially vibrating probe to detect the presence of light icing conditions. Goodrich ice detector is designed for thin ice layer applications like avionics and to the date no reports of detecting the ice under the water phase presence has been released.

HoloOptics T42 [6] employs IR signal passed through the medium and an external rain detector to eliminate the sources of false indications by water phase. No testing in marine icing conditions was conducted with the T42.

The Ice Meister Model 9734-SYSTEM industrial ice detector [7] monitors opacity and optical refraction of the ice along the contact with the probe surface. It only recognizes whether air, water or ice is present. This concept is somewhat similar to chilled mirror dew point sensor which often employs the optical reflectivity.

IDS-20 system [8] measures the complex impedance of the icing medium using capacitive plates hermetically sealed. The sensor can distinguish between water, however not in a combined multiphase state.

Jeung Sang Go with Xiang Zhi [9] employed capacitive sensors to measure ice growth in real time. The developed system has also been patented [10], however the authors conclude the water layer formation has to be prevented in order to maintain the accuracy. Charles Ezeoru [11] conducted a similar research using the same capacitive technique with interdigitated comb-style electrodes. Both approaches experienced the transformation from liquid state to solid state which has been reflected in a ramp capacitance profile in time. Unfortunately, no one has tried to quantify the transitional multiphase period.

We have developed a sensor array that allows a flexibility to be applied on curved surfaces such as wind turbine wings. Our earlier work described in [12] utilized a planar array of electrostatic sensors of different spacing that have been instrumented by a custom circuitry based on LC oscillator. Here we report on using a constant-spacing electrostatic sensor array interfaced and conditioned by a commercial four-channel capacitive pickup board by Texas Instruments.

Previously we processed the array signals by a multidimensional least squares method [12] and in another approach by artificial neural networks [13]. In this work, we applied a machine learning strategy centered around a decision tree method.

Machine learning algorithms are being used in many applications ranging from flood prediction [14], solar radiation [15], to wind generator blade monitoring [16,17]. Machine learning based on a decision tree method has been used in internet security systems [18] as well as in detecting stability of a power system voltage [19] or in classification tasks [20,21].

The ice can take different formations described in [22], however, the dielectric properties remain nearly constant which is being exploited in this research work. The problem of marine icing detection is however complicated through the presence of a water phase on top of the ice layer since water’s dielectric constant, the relative permittivity, is significantly larger than that of ice. The decision tree method has been found to meet our needs in recognizing and classifying the multiphase situations and provide a more accurate ice accretion measurement. This method translates the acquired signal data to a regression and classification model to determine the thickness of both layers of ice and water simultaneously.

## 2. Methodology

Our research into sensing and detecting the marine icing as a two phase phenomena has been initially founded on the principles of linearly decoupled array of electrostatic sensors of variable electrode gap spacing. The experimental verification confirmed our hypothesis of dissimilar gap spacing being capable to encode the stray electrostatic field above the sensor plane in a unique and linearly independent way. This phenomena was utilized in discriminating each individual phase, ice and water. Our team developed a multi-dimensional least-squares-method [12] to map the measured data to the measurands. The least-squares method was later substituted by a more robust and more accurate technique based on artificial neural networks method [13].

In this paper, we modified our earlier methods in four ways, (1) at the transducer array level (*constant gap spacing, variable insulator height*), (2) at the signal pick up circuitry (*commercial 4ch converter*), (3) at the signal processing algorithm level (*decision tree method*) and (4) at the spacial adaptation to curved surfaces (*structural flexibility*). Our earlier setup resulted in a large variation of capacitance range which required a custom build circuitry for capacitance to frequency conversion. In this work, we adapted an off-the-shelf signal detection board FDC2214EVM from Texas Instruments [23] which features four independent channels simultaneously scanned and a simple interfacing (Figure 1). FDC2214EVM module features 4-ch 28 bit capacitance-to-digital signal converters (FDC). The manufacturer also provides a software enabling the data streaming into a PC. In order to keep all array elements within a consistent accuracy range, we have built a new three element array of uniform elements (uniform spacing) that was attached to a curved surface of 80 mm radius (Figure 2).

In order to linearly decouple the measured capacitance across the array, we introduced three different dielectric layer heights. The capacitance of two elements was analyzed by a finite element method (FEM) using Ansys Maxwell software (Figure 3). The solid line characteristics represent the capacitance for an element having 0.25 mm coating of PET, plotted as constant capacitance contours across a range of ice and water accretion levels. The dotted line corresponds to 0.35 mm dielectric coating layer. The graph demonstrates a linear independence in a similar way to the gap variation analysis in [12] though not as profoundly. The FEM parameters are listed in Table 1.

Any traditional insulation material that is not permeable to water will provide the function of eliminating the conductance (real part) from the impedance measurement. PET sheets were readily available to us and their dielectric constant is similar to the epoxy resin used in the past. The critical part in obtaining the linearly independent characteristics across the array is in a different number of stacked layers of PET sheets above each array element.

## 3. Experimental Validation

We have conducted a series of ice layer deposition experiments in a similar way described in [12] using a water saturated roller. The system was placed in a deep freezer at −20 ${}^{{}^{\circ}}$C and was initially tempered for 1 h. Then, over a period of two hours, we have periodically rolled a water layer onto the sensor surface which is illustrated in Figure 4. Each peak corresponds to a new wetting cycle during which the capacitance rises sharply. If we were to consider only the ice layer as a number of previously developed sensors were based upon, the reported ice layer accretion would follow the same sharp increase causing a large discrepancy from the real situation. A low pass filter could be applied to remedy these time limited deviations however the water phase could also be present at all times in situations of icy waters constantly battering the place of interest.

Figure 4 shows six transitional cycles where each cycle starts with a fresh water layer that gradually solidifies into a new ice layer. Each cycle took approximately fifteen minutes to settle. Our experimental approach is based on an assumption that each deposited layer is consistent with all the other layers without the need for knowing its exact dimensions which constitutes a parametric approach.

## 4. Signal Processing

The Decision tree model is constructed from the main root branches consisting of the interior nodes and the final nodes also called the decision nodes [24]. The tree represents a mapping characteristic between the signal data and the target parameter, the ice accretion besides the water phase information as a byproduct.

We have split the acquired signal data and the corresponding ice and water accretion estimate into the training data set and the test data set. The training data is used to derive the decision tree parameters during the training phase. Figure 5 shows both the training and the testing processes. The training data set consists of 50% of data points from all the acquired data (odd rows in a column vector). Once the relationship between input data and the measurand was established, the testing phase has been conducted on the test data (even rows). Figure 6 shows the ’pairplot’ data set illustrating the relationship and its weight between the sensor data and the output classes.

Our decision tree implementation is based on the Entropy rule described in [25]. In comparison to other models like the Gini rule, the entropy model performed better in our specific application. Both rules are characterized in Equation (Equation 1) with $p_{i}$ being the entropy measure.
(1)$$\begin{array}{c} \mathrm{Gini}=1-\sum_{i=1}^{c}{\left(p_{i}\right)}^{2}\\ \mathrm{Entropy}=\sum_{i=1}^{c}-p_{i}log_{2}\left(p_{i}\right) \end{array}$$

Figure 7 shows our assumption in the time profile of our anticipated data of ice accretion (dotted line) and the water level height (solid line) [12]. At the end of each cycle, the water layer is completely converted into the ice.

## 5. Results

We implemented a twelve level deep decision tree processes the measured data based on entropy value where a low entropy value leads to the best leaf node in terms of purity. Applying the trained decision tree model to the test data results in the ice accretion estimate depicted in Figure 8 and the water layer estimate depicted in Figure 9. Both profiles show a significant improvement over the earlier results reported in [12] and a corresponding results to the ANN approach in terms of the ice accretion [13].

The Precision, Recall, F-Measure and Accuracy are commonly used for evaluating classification methods in Machine learning [26]. Precision is the ratio between the correct positive predicted classes for a layer and the total number that includes the correct and incorrect positive classes predicted for the same layer. Recall is the ratio between the correct positive predicted classes for a layer and the total number of the correct and incorrect negative classes predicted for the same layer. F-Measure makes a harmonic relation between Precision and Recall. These parameters were calculated as follows:(2)$$\begin{array}{c} \mathrm{Precision}=\frac{TP}{TP+FP}\\ \mathrm{Recall}=\frac{TP}{TP+FN}\\ F-\mathrm{Measure}=\frac{2\times \mathrm{Precision}\times \mathrm{Recall}}{\mathrm{Precision}+\mathrm{Recall}} \end{array}$$

True Positive (*TP*) is a classification method outcome where the decision tree method generates a correct positive class layer. Likewise, True Negative (*TN*) is a classification method outcome where the decision tree method generates a correct negative class layer. False Positive (*FP*) and False Negative (*FP*) are a classification method outcome where the decision tree method generates incorrect positive and negative class, respectively.

The Confusion-matrix yields the most ideal suite of metrics for evaluating the performance of a classification algorithm. Gupta [27] provides a detailed description on the confusion matrix as a measure of determining the accuracy of the decision tree classification model. The confusion matrix for ice and water classifier is a 6 × 6 matrix illustrated in Figure 10 with the elements along the diagonal representing ’True Predicted’ data. The overall accuracy for the prediction model of Ice and Water Classifier recorded accuracy of 92.8%.

The conventional method of using the least squares method to our experimental data as reported in [12] is presented in Figure 11 and Figure 12. Comparing these results with the decision tree results clearly indicate the level of improvement by using the new signal processing method. A comparison with the ANN method applied to the same data (Figure 13 and Figure 14) reveals a similar error characteristic in the ice accretion. The comparison analysis is quantitatively summarized by a root mean squares error analysis presented in Table 2. Both the decision tree and the ANN methods are equivalent in terms of the ice accretion measurement estimation, however, the current tree approach offers more lightweight implementation suitable for a microcontroller based system.

## 6. Conclusions

A novel sensor array has been developed to measure the marine icing accretion and to allow flexibility in deployment over curved surfaces. A new signal processing method based on a decision tree method has been applied to the array data. A theoretical analysis confirmed linearly decoupled signal characteristics that are critical in discriminating the water phase influence on the ice accretion measurement. An experimental analysis validated the high relevance of employing the decision tree method for signal processing. The ice accretion estimate obtained through the decision tree method demonstrates a significant improvement over the conventional least square method and a similar characteristic with the Neural Network method, however of a lesser computational overhead and a smaller footprint. Our three element capacitive array is easy to fabricate with off-the-shelf components and the 4 ch 28 bit signal processing interface circuit from TI is also aligned with a very low cost characteristics. The nature of our non-contact based design without any moving parts and the low cost characteristics promises a robust solution towards the marine icing measurement under the harsh environmental conditions.

## Figures and Tables

**Figure 1 sensors-21-00612-f001:**
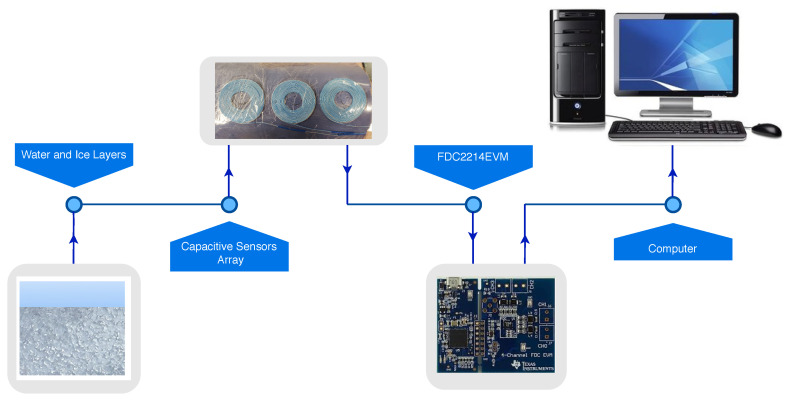
Interfacing Curved Capacitive Sensors Array with Computer by using FDC2214EVM.

**Figure 2 sensors-21-00612-f002:**
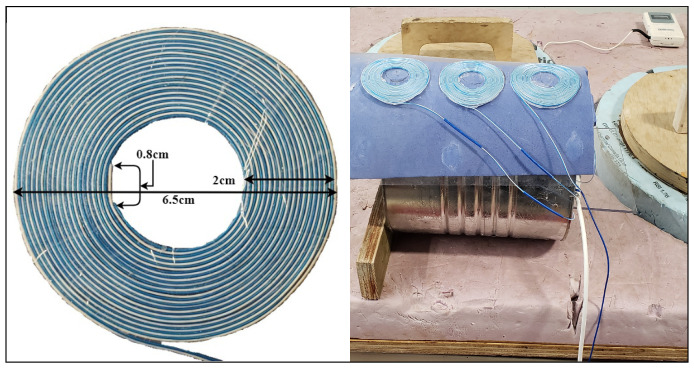
Spiral Capacitive Sensor Element.

**Figure 3 sensors-21-00612-f003:**
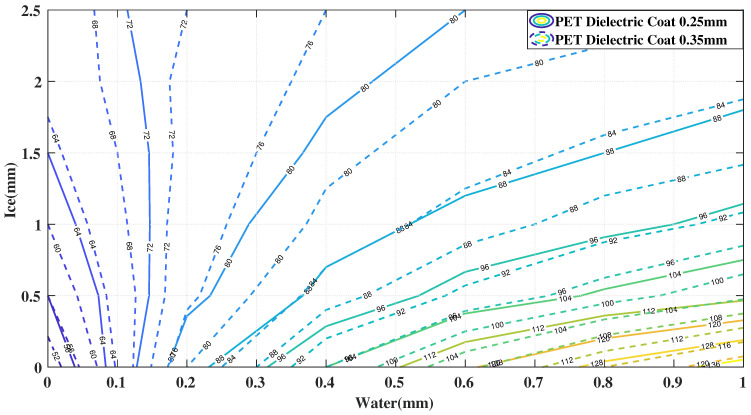
Contours of Constant Capacitance for Sensor Geometries 0.25 mm and 0.35 mm.

**Figure 4 sensors-21-00612-f004:**
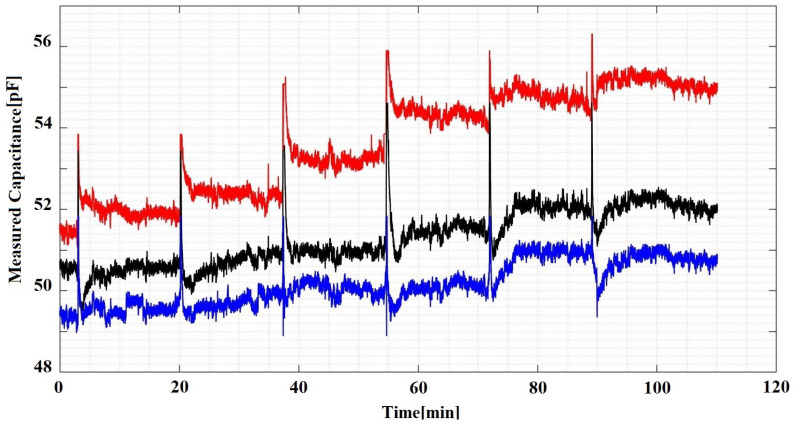
Experimental Data from Curved Capacitive Sensors Array.

**Figure 5 sensors-21-00612-f005:**
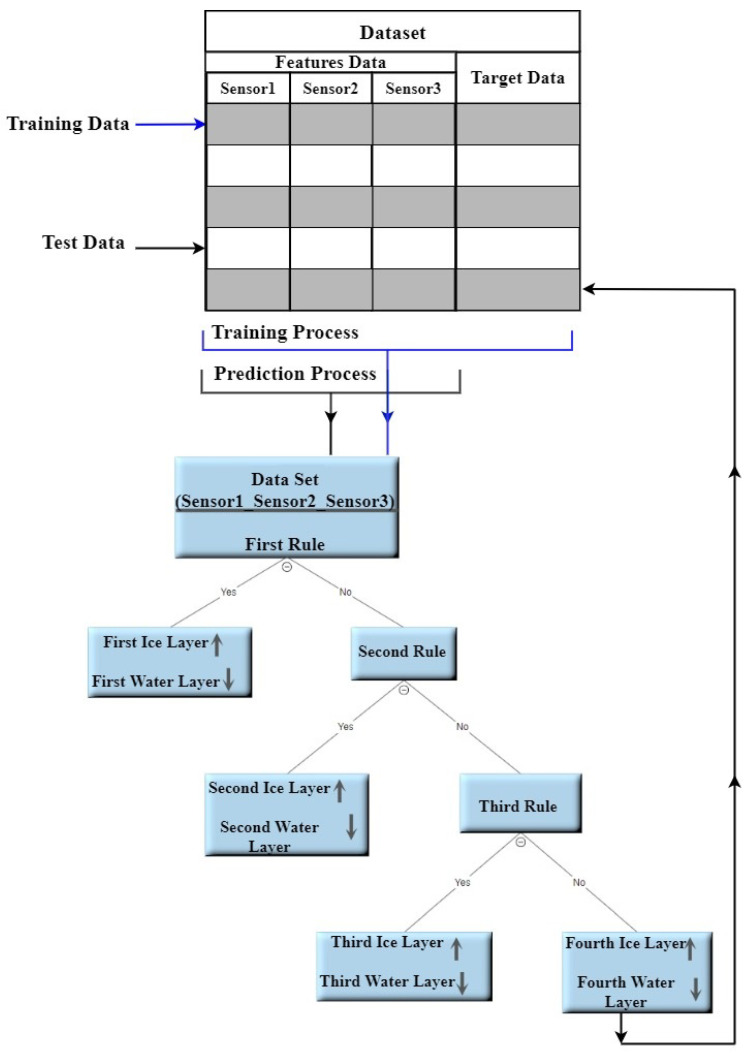
The Decision Tree Processes.

**Figure 6 sensors-21-00612-f006:**
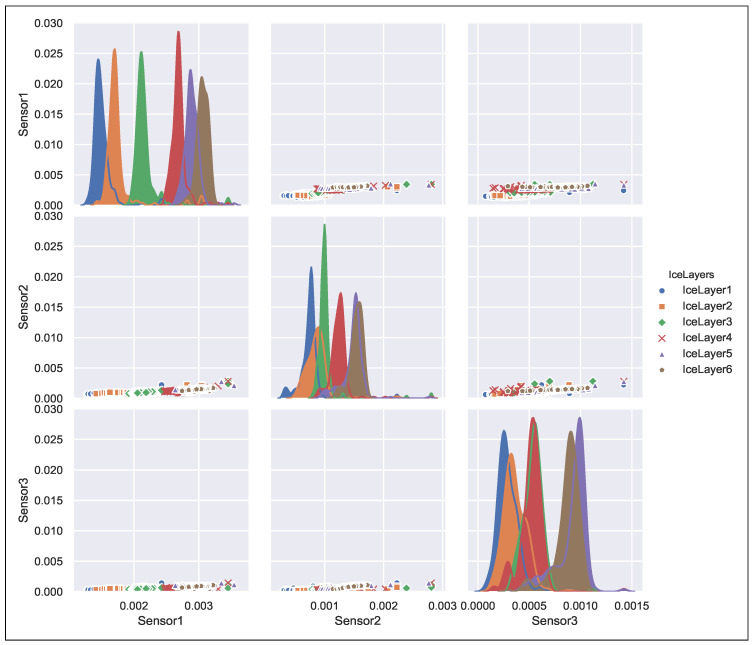
Pairplot data set for curved capacitive sensors array.

**Figure 7 sensors-21-00612-f007:**
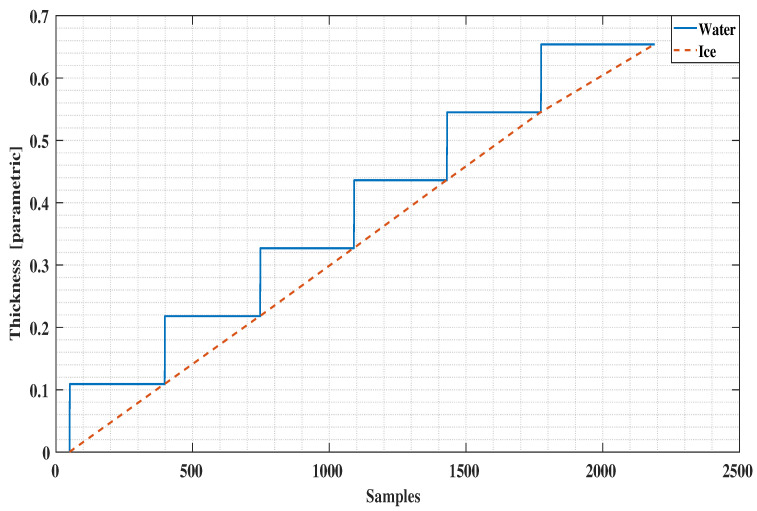
Ice and Water accretion in time during the experiment.

**Figure 8 sensors-21-00612-f008:**
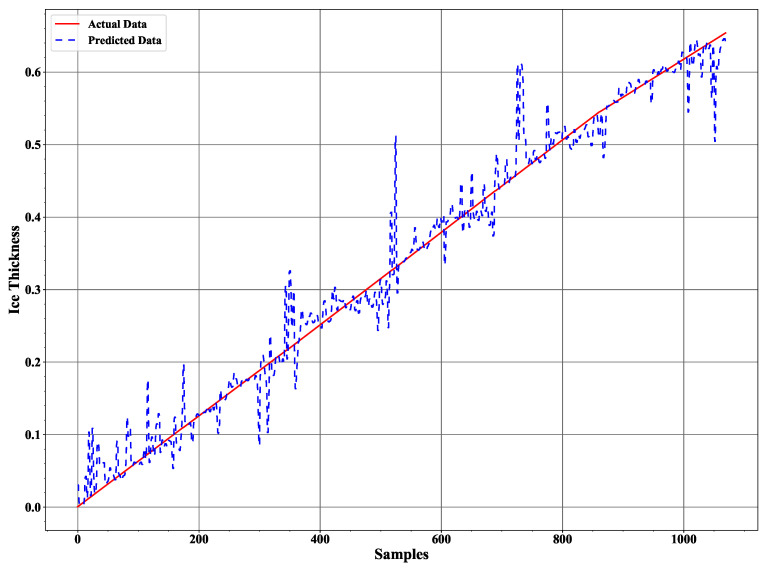
The ice accretion estimate in comparison to the assumed data characteristic.

**Figure 9 sensors-21-00612-f009:**
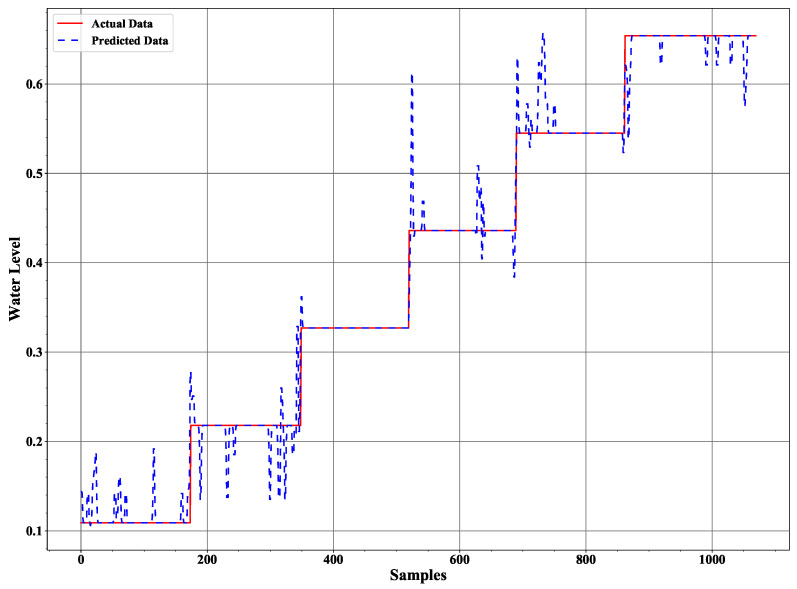
The water layer estimate in comparison to the assumed data characteristic.

**Figure 10 sensors-21-00612-f010:**
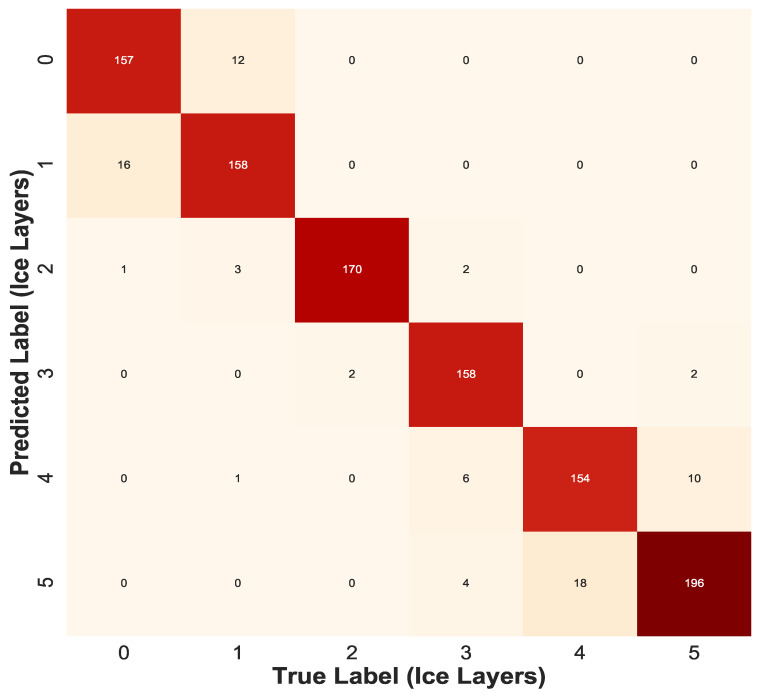
Confusion Matrix for Decision Tree Ice Classifier.

**Figure 11 sensors-21-00612-f011:**
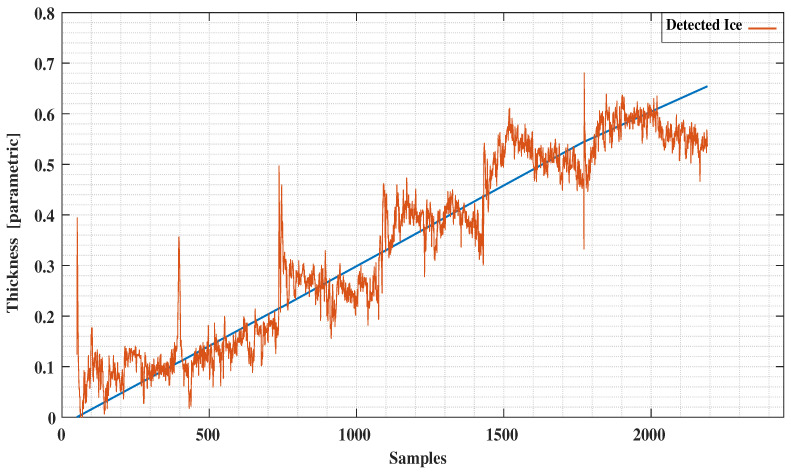
Detected Ice vs. Predictions using Least Squares.

**Figure 12 sensors-21-00612-f012:**
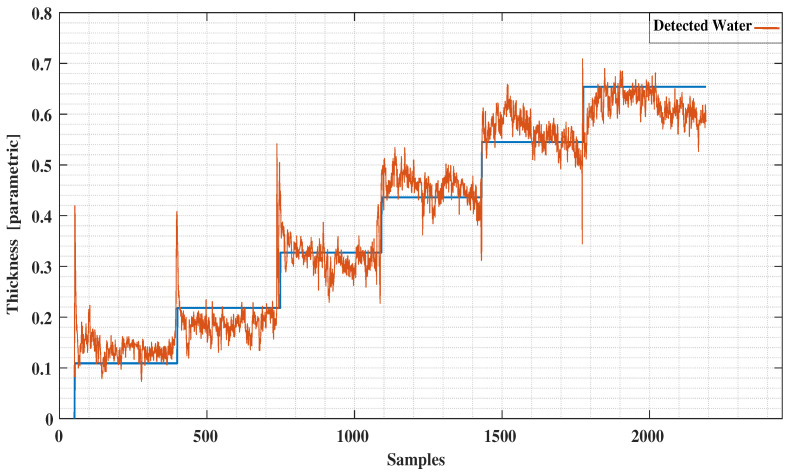
Detected Water vs. Predictions using Least Squares.

**Figure 13 sensors-21-00612-f013:**
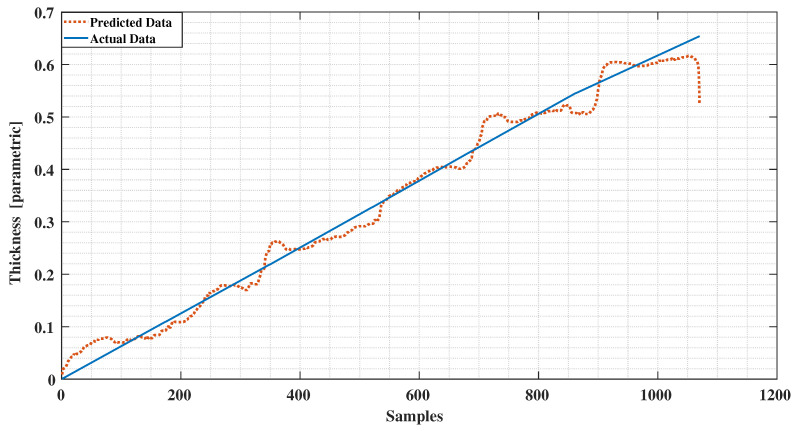
Detected Ice vs. Predictions using Neural Network.

**Figure 14 sensors-21-00612-f014:**
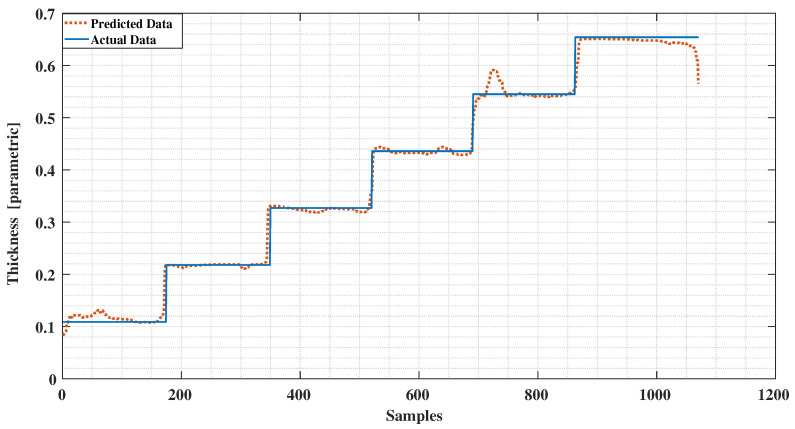
Detected Water vs. Predictions using Neural Network.

**Table 1 sensors-21-00612-t001:** FEM Model Parameters.

Material	$\varepsilon_{r}$
air	1.0006
water	81
ice	4.2
PET	3.6

**Table 2 sensors-21-00612-t002:** Error Analysis.

Method	Medium	Root Mean Square Error	Mean Absolute Error
Decision Tree	water	0.0231	0.0082
	ice	0.0291	0.01704
Least Squares [12]	water	0.0453	0.0336
	ice	0.0556	0.0424
Neural Network [13]	water	0.0144	0.0072
	ice	0.0214	0.01702

## Data Availability

The MATLAB and Python code introduced in the paper can be found at https://sciprofiles.com/profile/1438257.

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
