# Peer review of "Marine Icing Sensor with Phase Discrimination"

_sensors, 2021, doi:10.3390/s21020612_

Round 1
Reviewer 1 Report
PET means polyethylene terephthalate?, if so, I suggest to explain why PET has the appropriate dielectric propeties for these experiments
Numbers in Fig 6 are too small
How the authors relate dielectric properties with ice thickness? is there any physical equation?
Author Response
Thank you for reviewing the manuscript. Please see the attachment.

Reviewer 2 Report
Form Introduction does not result clearly which is the state-of-the-art method for icing detection.
Line 54 - please cite a technical documentation for FDC2214EVM board from TI, such as its user guide.
Line 60 – please give more details about the FEM simulation performed in Ansys Maxwell.
In section 2 please give a short description of the cited research from [3], and about the previous methods used [4]. Also, please include a detailed description of the current methodology that comprises of three variants of the approach.
In section 3 please give some details about the dielectric properties of ice versus liquid water.
The data sets (used as training data sets and test data sets – line 86) were experimentally obtained, inherited from a database or automatic/manually generated?
Figure 5 – please remove the filename `data2.jpg`.
Figure 8 and 9 – please remove the filename `Regression.pdf`.
In section 5 would be nice to have a description of the proposed method data processing and validation in comparison with the cited methods [3], [4].
I also expected to see the three ways of modification (line 50), I identify only a version. Also, if only one approach is described, please state that is the best of them and the other two should also be described briefly.
Conclusion section should be extended and some practical aspects, such as cost, or easiness of application, should be discussed. Why this approach is better than state-of-the-art methods? Are there some drawbacks? Could be easily implemented in wind farms and gas platforms with the existing equipment?
I think the entire paper should be read carefully and some additions should be made for clarity and to be easily understandable by the reader.
Author Response
Comments and Suggestions for Authors
- Form Introduction does not result clearly which is the state-of-the-art method for icing detection.
The introduction section was expanded with additional information about the state-of-the-art for icing detection.
- Line 54 - please cite a technical documentation for FDC2214EVM board from TI, such as its user guide.
Information about the interface board FDC2214EVM is now included.
- Line 60 – please give more details about the FEM simulation performed in Ansys Maxwell.
The referenced publication [3] describes the FEM simulation of this problem in a more detail. The publisher recommended this work to be published in 'Letters' and therefore we focused more on the results. A table of FEM parameters has been added.
- In section 2 please give a short description of the cited research from [3], and about the previous methods used [4]. Also, please include a detailed description of the current methodology that comprises of three variants of the approach.
The section 2 has been expanded accordingly to your suggestion including a detailed description of references [3,4] and more detail is also provided towards the current approach.
- In section 3 please give some details about the dielectric properties of ice versus liquid water.
The FEM parameters table added as per the above request lists the requested parameters.
- The data sets (used as training data sets and test data sets – line 86) were experimentally obtained, inherited from a database or automatic/manually generated?
In this work we only used the measured data from an experimental validation work. ODD vectors have been used for training while EVEN vectors have been used in the testing phase.
- Figure 5 – please remove the filename `data2.jpg`.
The images are now presented properly. Thank you.
- Figure 8 and 9 – please remove the filename `Regression.pdf`.
The images are now presented properly. Thank you.
- In section 5 would be nice to have a description of the proposed method data processing and validation in comparison with the cited methods [3], [4].
This section has been expanded with additional comparisson between the current work and [3,4].
- I also expected to see the three ways of modification (line 50), I identify only a version. Also, if only one approach is described, please state that is the best of them and the other two should also be described briefly.
The Section 2 is expanded with an additional description about the current approach and its modification from the previous work referenced in [3,4].
- Conclusion section should be extended and some practical aspects, such as cost, or easiness of application, should be discussed. Why this approach is better than state-of-the-art methods? Are there some drawbacks? Could be easily implemented in wind farms and gas platforms with the existing equipment?
The Conclusion section has been expanded and some practical aspects added.
Reviewer 3 Report
Major comment:
There is no discussion of results in this manuscript. In section 5, the results are shown in Figures from 8 to 12 and superficially compared to previous works. A detailed discussion of results is fundamental to make clear their relevance and novelty.
Minor comments:
- On the Figure 2 caption, I suggest to replace "capacitor" by "capacitive";
- Although the spiral capacitive sensor and its dimensions are shown in Figure 2, in methodology or in experimental validation section, the technical parameters of this sensor can be provided with more details;
- I think this manuscript is not a Letter article type. It has 11 pages, 12 figures, 16 references and more than 2600 words of text. Therefore, it could not be be considered a brief disclosure of original research. In this respect, I suggest to the authors to consider extending the text in order to submit the revised version as original research article. This could be made by detailing the research background in introduction and detailing the dicussion of results.
Author Response
Major comment:
- There is no discussion of results in this manuscript. In section 5, the results are shown in Figures from 8 to 12 and superficially compared to previous works. A detailed discussion of results is fundamental to make clear their relevance and novelty.
The Section 5 has been expanded with an additional comparisson to the state-of-the-art methods. Thank you.
Minor comments:
- On the Figure 2 caption, I suggest to replace "capacitor" by "capacitive";
Has been replaced as per your suggestion.
- Although the spiral capacitive sensor and its dimensions are shown in Figure 2, in methodology or in experimental validation section, the technical parameters of this sensor can be provided with more details;
All three array elements are of the same construction with the only difference of insulator dielectric layer thickness (PET foil). Additional design parameters of the array element has been included.
- I think this manuscript is not a Letter article type. It has 11 pages, 12 figures, 16 references and more than 2600 words of text. Therefore, it could not be considered a brief disclosure of original research. In this respect, I suggest to the authors to consider extending the text in order to submit the revised version as original research article. This could be made by detailing the research background in introduction and detailing the dicussion of results.
This publishing format has been recommended by the publisher upon submission.
Reviewer 4 Report
The paper reports a study of an electronic circuit for the measurement of marine icing phenomena.
General comment:
The paper presents a measurement apparatus for the icing phenomena. The letter is well written, fluent and easy to follow. The figures instead should be better accomplished with captions and description. I have some questions and comments that I suggest to be addressed to better complete the description of the concept.
1) Is the sensor array mentioned in the letter to be considered an individual set of three sensors or it can be extended to larger set of sensors? or a larger array composed of sets of three sensors ?
3) Can you specify the values of the capacitances? Are these in the range of pF, nF or what? and how much do the vary depending of the ice thickness?
4) NormCap in figure 6 is what? a normalized capacitance? in this case, normalized to what?
3) Does the length of the wires to link the sensor affect the capacitance? In this case it is just an additional capacitance to be considered or the entire NN must be trained depending on the experimental setup?
4) Please in figures 6 be more precise in the captions and the description of the axes. The numbers refer to what?
Specific comments
- line 24: Colpitts --> I am wondering if it is worth saying that the oscillator is a Colpitts, I suggest just to mention the oscillator circuit
- line 32: IoT --> please specify the acronym Internet of Things, I guess?
-
I recommend to minor revise the manuscript by addressing the comments before accepting it for the publication
Author Response
General comment:
The paper presents a measurement apparatus for the icing phenomena. The letter is well written, fluent and easy to follow. The figures instead should be better accomplished with captions and description. I have some questions and comments that I suggest to be addressed to better complete the description of the concept.
1) Is the sensor array mentioned in the letter to be considered an individual set of three sensors or it can be extended to larger set of sensors? or a larger array composed of sets of three sensors?
The array of three sensing elements of slightly varying characteristic has been chosen to provide a redundancy for the two-phase phenomena detection. Two parameters can be independently extracted by two dissimilar sensors similarly to two linearly independent equations required for two unknown variables. The extra sensing element is employed for the redundancy reasons.
3) Can you specify the values of the capacitances? Are these in the range of pF, nF or what? and how much do the vary depending of the ice thickness?
Newly updated Figure 6 provides the data magnitudes as per your next item below.
4) NormCap in figure 6 is what? a normalized capacitance? in this case, normalized to what?
The capacitance normalization was introduced in our previous research to fit all characteristics onto one linear scale graph window [3] using [(data-shift)/scale] which still preserves the linear independence. In this work, we do not however normalize the data. A proper formatting of the graph has been uploaded. Thank you for spotting this omission.
3) Does the length of the wires to link the sensor affect the capacitance? In this case it is just an additional capacitance to be considered or the entire NN must be trained depending on the experimental setup?
The sensor-to-interface connection does introduce a constant parasitic capacitance in parallel with the main sensor however due to its constant nature besides the short wire length the resulting small bias (shift) has no effect towards the linear independence of the array elements characteristics.
4) Please in figures 6 be more precise in the captions and the description of the axes. The numbers refer to what?
The Figure 6 has been updated as per your comments.
Specific comments
- line 24: Colpitts --> I am wondering if it is worth saying that the oscillator is a Colpitts, I suggest just to mention the oscillator circuit.
Updated
- line 32: IoT --> please specify the acronym Internet of Things, I guess?
Updated
Round 2
Reviewer 1 Report
The manuscript was improved as suggested, it is ready for its publication
Reviewer 2 Report
Thank you for your collaboration!
I want only to ask you to perform some small text and figure arrangement and editing.